Comprehensive analyses of genomes, transcriptomes and metabolites of neem tree

Kuravadi Nagesh A. 1
Yenagi Vijay 1
Rangiah Kannan 2
Mahesh HB 1 3
Rajamani Anantharamanan 1
Shirke Meghana D. 1
Russiachand Heikham 1
Loganathan Ramya Malarini 1
Shankara Lingu Chandana 1
Siddappa Shilpa 1
Ramamurthy Aishwarya 1
Sathyanarayana BN 4
Gowda Malali 1 malalig@ncbs.res.in malalig@ccamp.res.in
1 Genomics Laboratory, Centre for Cellular and Molecular Platforms, National Centre for Biological Sciences , Bangalore, Karnataka , India
2 Metabolomics Facility, Centre for Cellular and Molecular Platforms, National Centre for Biological Sciences , Bangalore, Karnataka , India
3 Marker Assisted Selection Laboratory, Department of Genetics and Plant Breeding, University of Agricultural Sciences, GKVK , Bangalore, Karnataka , India
4 Plant Tissue Culture Laboratory, University of Agricultural Sciences, GKVK , Bangalore, Karnataka , India
Kumar Abhishek
Electronic publication date: 2015 Aug 6
Publication date: 2015
Volume: 3
Electronic Location ID: e1066
Received 2015 Jan 25; Accepted 2015 Jun 9
Copyright: © 2015 Kuravadi et al.
Copyright year: 2015
Copyright holder: Kuravadi et al.
License: This is an open access article distributed under the terms of the Creative Commons Attribution License, which permits unrestricted use, distribution, reproduction and adaptation in any medium and for any purpose provided that it is properly attributed. For attribution, the original author(s), title, publication source (PeerJ) and either DOI or URL of the article must be cited.
License URL: https://creativecommons.org/licenses/by/4.0/

Keywords: Next generation sequencing, Annotation, A. indica, Azadirachtin

Funding: Department of Biotechnology, Government of India (Ramalingaswami Fellowship Grant) BT/HRD/35/02/2006 This work was supported by Department of Biotechnology, Government of India to Malali Gowda (Ramalingaswami Fellowship Grant; BT/HRD/35/02/2006). The funders had no role in study design, data collection and analysis, decision to publish, or preparation of the manuscript.

==============================
Neem (Azadirachta indica A. Juss) is one of the most versatile tropical evergreen tree species known in India since the Vedic period (1500 BC–600 BC). Neem tree is a rich source of limonoids, having a wide spectrum of activity against insect pests and microbial pathogens. Complex tetranortriterpenoids such as azadirachtin, salanin and nimbin are the major active principles isolated from neem seed. Absolutely nothing is known about the biochemical pathways of these metabolites in neem tree. To identify genes and pathways in neem, we sequenced neem genomes and transcriptomes using next generation sequencing technologies. Assembly of Illumina and 454 sequencing reads resulted in 267 Mb, which accounts for 70% of estimated size of neem genome. We predicted 44,495 genes in the neem genome, of which 32,278 genes were expressed in neem tissues. Neem genome consists about 32.5% (87 Mb) of repetitive DNA elements. Neem tree is phylogenetically related to citrus, Citrus sinensis. Comparative analysis anchored 62% (161 Mb) of assembled neem genomic contigs onto citrus chromomes. Ultrahigh performance liquid chromatography-mass spectrometry-selected reaction monitoring (UHPLC-MS/SRM) method was used to quantify azadirachtin, nimbin, and salanin from neem tissues. Weighted Correlation Network Analysis (WCGNA) of expressed genes and metabolites resulted in identification of possible candidate genes involved in azadirachtin biosynthesis pathway. This study provides genomic, transcriptomic and quantity of top three neem metabolites resource, which will accelerate basic research in neem to understand biochemical pathways.

Introduction

Neem, Azadirachta indica is an evergreen tree, native to the Indian subcontinent. It belongs to Meliaceae family plants, which are the major source for diverse limonoids (Tan & Luo, 2011). Neem has been used in Ayurveda, Siddha, Unani and other Indian local health traditions. Over 700 herbal preparations and over 160 local practices containing neem ingredients are known in India, which claim to prevent various ailments or disorders in humans (Brahmachari, 2004). Neem based pesticidal formulations are widely regarded as organic, and are found to have low toxicity against non-target beneficial organisms as compared to synthetic pesticides. Nimbin was the first chemical limonoid isolated from neem tree (Siddiqui, 1942). Subsequently, more than 150 bioactive chemical compounds have been isolated from various neem tissues (Brahmachari, 2004).

Azadirachtin is the major tetranortriterpenoid in neem seeds (Butterworth & Morgan, 1968) and its molecular structure elucidation took more than 20 years of research (Broughton et al., 1986). Azadirachtin is one of the highly successful biopesticides in the world, which is isolated from neem seeds and is non-persistent in the environment. Its content is highly variable in trees in various locations due to genetic variability and environmental factors (Sidhu, Kumar & Behl, 2003). Chemical in-vitro synthesis of azadirachtin has been tried in the laboratory. However, long synthesis process and molecular complexity was realized and that chemical synthesis of azadirachtin is not a viable method for commercial production (Veitch, Boyer & Ley, 2008). Hence, there is an increased interest to understand the in-vivo biosynthesis of azadirachtin pathway in neem. Genes and proteins involved in biochemical azadirachtin pathways have not been researched in neem. Recently, a few studies were attempted to generate genomic resources for neem tree (Rajakani et al., 2014; Narnoliya et al., 2014; Krishnan et al., 2012). However, they generated limited ESTs (Rajakani et al., 2014; Narnoliya et al., 2014) and non availability of neem genome annotations (Krishnan et al., 2012). In this study, we have sequenced three neem tree genomes and transcriptomes using next generation sequencing technologies. In addition, we quantified neem transcripts and metabolites using RNAseq and UHPLC-MS/SRM methods, respectively. This work will accelerate research to dissect biochemical limonoid pathways in neem.

Materials and Methods

Neem genotypes

Individual neem tree was identified from three varied geographical regions of southern India including Karnataka (GKVK, Bangalore, India (abbreviated as Genotype 1), Anuganalu, Hassan (Genotype 2)) and Tamil Nadu (Erode; Genotype 3) for this study (Fig. S1).

Isolation of genomic DNA and total RNA from neem tissues

Mature neem leaves were collected from neem trees for DNA isolation. CTAB method (6% CTAB, 1.4 M NaCl, 20 mM EDTA, 10 mM Tris Base pH8) was used for isolation of neem genomic DNA (Doyle, 1990), with few modifications. Neem DNA isolation is a difficult process due to the presence of highly oxidized complex compounds in neem leaves. Instead of precipitating the DNA using iso-propanol as in CTAB method, we used the supernatant to purify genomic DNA using the Sigma Genelute plant DNA isolation kit (G2N70; Sigma, Seelze, Germany). RNA isolation was carried out using 100 mg of the neem tissues using Sigma Spectrum Plant Total RNA Kit (STRN50; Sigma, Seelze, Germany). DNase treatment was done using DNase digestion kit (DNASE 70; Sigma, Seelze, Germany). DNA free RNA was dissolved in DEPC water. The RNA integrity (RIN) was determined using Agilent Bioanalyzer.

Induction of callus tissue

Neem endosperm explants, of 4–5 mm length pieces were cultured on Murashige and Skoog solid medium (pH 5.6) containing 0.45 mg BAP, 0.80 mg NAA and 150 mg Casein hydrolysate. Cultures were maintained in growth chamber at 25 ± 2 °C with 55 ± 5% of relative humidity, for 16 h photoperiod and allowed for formation of callus tissue. After 2 weeks, mature callus was sub-cultured to fresh media every fourth or sixth weeks.

NGS library construction and sequencing

Whole genome shotgun DNA library was prepared using Illumina TrueSeq DNA sample preparation kit (FC-121-2001). The paired-end (PE) (2 × 100 nts) sequencing was carried out using Illumina HiSeq 1000 at the Next Generation Genomics Facility at Centre for Cellular and Molecular Platforms (C-CAMP). We prepared whole genome shotgun 454 library for Genotype 1 (GKVK, Bangalore, India) using the rapid library preparation kit from Roche (Cat. No. 05608228001y; version 4.0.12). The 454 sequencing was carried out using GS FLX+ chemistry as per Roche/454 manual instructions (http://454.com).

Transcriptome library preparation and sequencing

Total RNA isolation was carried out for 100 mg of the neem tissues using the Sigma Spectrum Plant Total RNA Kit (STRN50; Sigma, Seelze, Germany). The RNA-seq library was prepared using 1 µg of total RNA according to Illumina’s TruSeq RNA sample preparation kit (RS-122-2001).

De novo whole genome assembly

Illumina PE reads were pre-processed using FASTX-Toolkit (v 0.0.13). The read quality score cut-off (q) and percentage (p) value was assigned as 20 and 100, respectively (i.e., q/20 and p/100). After quality filtering, we obtained a total 75–192 millions of paired-end reads and 61–71 millions of singleton reads (Table S1). De novo assembly of neem genome was performed using Velvet program (Zerbino & Birney, 2008). We optimized the Velvet assembly through iteratively process for various k-mers (27 to 67 nts). The Velvet assemblies with best k-mers in the size of 45, 45 and 33 were used for Genotypes 1, 2 and 3, respectively. The parameter used for deciding the best k-mer for theoretical coverage were N50, maximum contigs length, totals contigs, assembled genome size and total number of reads used (Table S1).

Table 1 Neem genome assembly statistics.

Assembly parameters	Velvet assembly from Illumina reads	MIRA assembly from 454 reads	Hybrid assembly of Genotype 1	
Total high quality reads	168,895,379	2,762,254	–	
k-mer	45	–	–	
Assembled genome size (Mb)	216	157	268	
Total number of contigs	94,780	1,21,184	68,604	
N50 (bp)	22,263	1,463	15,948	
Maximum contig length (bp)	2,41,126	43,859	241,170	
Mininum contig length (bp)	89	52	89	
% of bases in contigs ≥ 1,000 bp	93.31	74.54	94.65	
Total repeat size in Mb (%)	59.81 (27.41)	48.84 (31.05)	86.90 (32.44)	
Number of predicted genes in Augustus	27,556	41,169	40,130	
Number of predicted genes in GeneScan	35,501	57,356	52,617	
Number of Genes clustered from GeneScan and Augustus	37,161	61,901	48,032	
No of genes with >100 bp	34,992	52,957	44,495	
Genes with RNA seq evidence	27,087	43,383	32,278	
Non-TE genes	19,547	41,373	29,050	

A total of 454 reads were assembled using MIRA software (Chevreux, 2005). The first step of the assembly is to compare every read with every other read (and its reversed complement) to detect potential overlaps. These potential overlaps were examined with Smith–Waterman-based algorithm for local alignment of overlaps. If overlaps were found, then they were verified using Smith-Waterman methods and are assembled into contigs. The software used default parameters (minimum read length of 40 nts and minimum base quality of q10) with single end read format.

Hybrid neem genome assembly generation using Illumina and 454 contigs

Hybrid assembly of neem genome was carried out by merging contigs from Illumina and 454 sequence reads. The Genotype 1 (GKVK, Bangalore, India) assembled contigs, obtained from Velvet (Zerbino & Birney, 2008) and MIRA (Chevreux, 2005) assemblers were merged using clustering program, CD-HIT-est by keeping minimum similarity cut-off of 90% (Li & Godzik, 2006). This clustering approach allowed both Illumina and 454 contigs to merge and build longer contigs of the genome. A total of 94,780 Illumina contigs were clustered with a total of 121,184,454 contigs, resulting in a total of 68,604 unique contig sequences (Table 1).

Eukaryotic core gene mapping

The completeness of the assembled genome was checked by using eukaryotic core gene-mapping approach (CEGMA) (Parra, Bradnam & Korf, 2007). CEGMA used 248 core eukaryotic genes that are highly conserved and single-copy genes in eukaryotic genomes.

Nuclear gene prediction and annotation

Genes were predicted from hybrid genome assembly using Augustus (Stanke et al., 2006) and GenScan (Burge & Karlin, 1997) (Table S2). Then we clustered genes with similarity cut-off 90% using CD-HIT-est program (Li & Godzik, 2006). This method gave the overall representation of genes, by merging similar genes predicted by both the programs. Then we discarded genes that were less than 100 bp in size (Delcher et al., 1999). Then BLAT was used to obtain the number of unique and common genes from the CD-HIT out file (Kent, 2002). Genes originated from repeat elements were identified using Repeat Modeler program (http://www.repeatmasker.org/RepeatModeler.html). Gene expression was quantified by mapping RNA-Seq reads (Table 2). We used all the genes with and without RNA-seq evidence to search gene functions using UniProt database, Gene Ontology (GO), Kyoto Encyclopedia of Genes and Genomes (KEGG) and Enzyme Commission number (EC). Schematic representations of GO classes in neem genome are summarized in Fig. S2. Above analyses was done using BLASTX (Altschul et al., 1990) in annot8r (Schmid & Blaxter, 2008) with E-value cut-off of 10−3. Genes with multiple hits were filtered based on E-value; the annotation details for each gene are listed in the Table S3.

Table 2 RNAseq analysis from various explants of neem.

Tissue	No. of readsa	No of genes/RPKM > 1	No of genes/RPKM > 5	No of genes/RPKM > 10	
Mature leaf	5,401,910	19,308	14,807	11,763	
Flower and bud	22,654,982	21,927	16,716	13,632	
Fruit coat and pulp	55,627,021	21,537	16,693	13,888	
Developing endosperm	31,340,522	19,480	15,262	12,614	
Mature fruit	23,321,657	17,366	12,407	9,741	
Seedling root	4,427,659	20,798	17,015	14,312	
Seedling shoot	5,894,621	20,199	15,926	13,018	
Drought root	7,255,199	21,371	17,015	14,236	
Drought shoot	22,600,138	20,763	16,077	13,431	
Albino root	1,267,4871	21,710	17,201	1,4273	
Albino shoot	23,115,676	21,226	16,874	14,066	
Leaf callus	2,150,935	18,615	15,356	12,727	
Notes.

a read quality = q20 and read length = 100 nts.

Gene expression analysis

The combined transcriptome of twelve tissues of neem tree from Genotype 1 were assembled using Trinity software (Grabherr et al., 2011). The transcripts were clustered to remove the over represented short fragments using CD-HIT-est program (Li & Godzik, 2006) with a minimum similarity cut-off of 90%. The 44,495 genes from CD-HIT-est were mapped with RNA-seq data from individual tissues using SeqMap (Jiang & Wong, 2008). The mapping of RNA-seq reads from each tissue were used to measure the expression value in RPKM (reads per kilo base per million) using rSeq tool (Jiang & Wong, 2009). The gene expression was estimated using RPKM value minimum ≥1 for further analysis. The RPKM value was used to cluster the genes according to their expression pattern using WCGNA package in R tool (Langfelder & Horvath, 2008). The expression value was also determined for assembled transcripts to verify expression of genes predicted from gene models.

Repeat prediction from neem genomes

De novo repeat identification was done using RepeatModeler (http://www.repeatmasker.org/RepeatModeler.html). The program was run with RM-BLAST (NCBI) database as an input for the repeat modelling. We trained Repeat Modeler using genomes of other published plant species including P. trichocarpa (39.40%), R. communis (51.74%) and A. thaliana (15.29%) for pipeline validation (Table S4). We downloaded NCBI data (SRA1085705) from Krishnan et al. (2012) and re-built the neem genome assembly using SOAPdenovo2 program with different k-mers (Table S5).

Quantitative PCR (qPCR) analysis

Total RNA from leaf, callus and developing endosperm (S1 =10 days post seed setting, S2 = 20 days post seed setting, S3 = 30 days post seed setting, S4 = 40 days post seed setting) was isolated using TRIzol reagent method and quantified using a Qubit Fluorometer. The cDNA synthesis was performed using total RNA (1 µg) with oligo(dT) random primers (50 µM) and SuperScript® III RT enzyme (200 u/µl) (Cat # 18080044; Life Technologies, Carlsbad, California, USA). The qPCR was performed on an Applied Biosystems, 7900HT Fast Real-Time PCR system machine. Real-time PCR was performed in a 384-wells optical reaction plate (Applied Biosystems, Foster City, California, USA) using SYBR green PCR mastermix (Life Technologies, Cat #4344463), which contains AmpliTaq Gold® DNA polymerase and ROX as a passive reference dye. Cycling conditions were 95 °C for 15 s, 60 °C for 30 s and 72 °C for 30 s with 40 cycles. We compared the fold change in qPCR experiments for selected genes from azadirachtin biosynthesis with conserved eukaryotic (rice) elongation factor 1α (eEF-1α) gene (AK061464.1). All reactions were performed in triplicate with elongation factor primers and water as an internal control. The primer sequences of selected genes are listed in the Table S6.

SSRs, SNPs and InDels analysis

Identification of simple sequence repeats (SSRs) or microsatellite was done using MIcroSAtellite tool (MISA) (http://pgrc.ipk-gatersleben.de/misa/) with assembled neem genome sequences. The SSRs containing contigs were extracted for SSRs motif variability prediction. The previously published SSR marker regions (Boontong, Pandey & Changtragoon, 2009) for neem were compared to the newly identified SSRs from the genome. The SSR based polymorphism among the neem accessions was done using an in-house software pipeline. This pipeline uses the identified SSR region along with 100 bp upstream and downstream from each SSR loci. The SSR regions were aligned to each other for a pair of genomes using Bowtie2 alignment. The resulting .SAM files were parsed using the libraries function of genomic ranges (Aboyoun, Pages & Lawrence, 2010), Gtools (Warnes, Bolker & Lumley, 2008) and Stringr (Wickham, 2010) in R program to obtain polymorphic SSR regions between neem genomes. The concordance was taken with neem tree Genotype 1 as a reference to shortlist the most polymorphic SSRs.

Single nucleotide polymorphism (SNP) and Insertion Deletion (InDels) markers were identified by mapping Illumina short reads from Genotype 2 and Genotype 3 to reference Genotype 1 assembly using Bowtie2 (Langmead & Salzberg, 2012). The alignment results were converted into .BAM format using Samtools v1.19 (Li et al., 2009). All .BAM files were merged into a combined .BAM file. The .BAM file was sorted, indexed and duplicates removed using Samtools v1.19 for further analyses. VCF file was generated for each genome to obtain SNPs and InDels from neem genome. The SNPs and InDels were further filtered to obtain aligned reads with quality >30, and minimum sequence depth of 10 reads. The SNPs were annotated using snpEff tool (Cingolani et al., 2012).

Neem chloroplast and mitochondrial genome assembly and annotation

The reads for chloroplast and mitochondria were extracted separately by mapping genome reads to chloroplast and mitochondrial genomes of known plants; A. thaliana, B. napus, C. papaya, C. sinensis, N. tabacum, P. dactylifera, P. trichocarpa, R. communis, S. bicolour and V. vinifera using Bowtie2 (Langmead & Salzberg, 2012). The mapped reads were extracted using Samtools (Li et al., 2009) and assembled separately using Velvet (Zerbino & Birney, 2008). Gene prediction and assembly of chloroplast genome was done using DOGMA (Wyman, Jansen & Boore, 2004). Gene prediction and annotation of the mitochondrial assembly was done using Mitofy (Alverson et al., 2010).

Synteny analysis

We aligned neem contigs on citrus (Citrus sinensis) chromosomes using MUMMER program (Kurtz et al., 2004). Synteny was computed among neem and citrus using Symap4.0 (Soderlund, Bomhoff & Nelson, 2011). The synteny information was visualized using the Perl script provided with Symap4.0.

Classification of gene families and phylogenetic analysis

The proteome of 23 sequenced plant species along with neem were used to search for homologues and unique genes. All-vs-all BLAST-P (E-value e-10) was done using Proteinortho program (Lechner et al., 2011). Comparative data from BLAST analysis was further classified into list of potential orthologs, co-orthologs and paralogs. Besides classifying, the program has also grouped the proteins into specific groups by clustering the gene-pairs. The gene content at the ancestral nodes along with the branches was reconstructed by using Wagner Parsimony and Likelihood-based approaches in the program count (Csűös, 2010).

Quantification of azadirachtin, salanin and nimbin using UHPLC-MS/SRM method

We purchased neem standard metabolites such as azadirachtin (Cat. No. A7430; Sigma-Aldrich, Madhya Pradesh, India), salanin (Cat. No. ASB-00019028-005; Chromadex, Irvine, California, USA) and nimbin (Cat. No. N476280; Toronto Research Chemicals, Toronto, Canada). High purity MS grade solvents (methanol, acetonitrile and water) were obtained from Merck Millipore (Merck Millipore India Pvt. Ltd., Mumbai, India).

For metabolites analysis, we collected samples from various neem tissues (mature fruit, developing endosperm, mature leaf, flower and bud, fruit coat and pulp and seedling shoot and root) and dried at 37 °C for two days. The samples were ground into fine powder using Pestle and Mortar, and stored at −80 °C until use. There were no commercially internal standards available for neem metabolites quantification; therefore, we used estrone-d4 (Steraloids Inc, Newport, Rhode Island, USA) as an internal standard to construct the standard curve and also for absolute quantification. The calibration curves for the quantification of all three neem metabolites were linear over a 64-fold concentration range (azadirachtin and salanin) and 73-fold concentration range (nimbin) with linear regression correlation coefficients ranging from 0.998 to 0.999 (Fig. 1D). Typical UHPLC-MS/SRM chromatogram profile for neem metabolites from standard compounds and from the seed extract are shown in Fig. 1C. All analytes showed single sharp peak in C-18 column.

Figure 1 Estimation of major neem metabolites from different tissues of neem.

(A) Neem tree, (B) structure of neem metabolites, (C) UHPLC-MS/SRM chromatogram of standards and metabolites from seed extract, (D) standard curve for all three metabolites and (E) concentration of neem metabolites from various tissues of neem tree.

Neem metabolites were extracted from the dried powder (2 mg) using 1 mL of methanol followed by 5 min sonication and centrifuged for 5 min (13,000 rpm, 10 °C), the supernatant was transferred to fresh micro-centrifuge tubes. About 5 µL of supernatant was spiked into 35 µL of methanol along with internal standard (10 µL estrone-d4 from 100 µg/mL). The analyses were done by injecting 10 µL of the sample into the UHPLC-MS/SRM system (LC-Agilent 1290 infinity series, MS–Thermo Fishers TSQ vantage). The intense product ions were selected for the LC-MS/SRM analysis [azadirachtin (703 → 567 Da), nimbin (541 → 509 Da), salanin (597 → 419 Da), and for estrone-d4 (275 → 257 Da)]. We used following LC conditions: solvent system A-Water (10 mM Ammonium Acetate) containing 0.1% FA and B-Acetonitrile containing 0.1% FA, Flow-200 µL/min, Column- C-18 column (Shim-pack, ODS III, 2.1 ×150 mm, 2 µm), Gradient- 2% B at 0 min, 2% B at 3 min, 40% B at 10 min, 95% B at 15 min, 2% B at 15.1 min, 2% B at 15.1–20 min. MS conditions: spray voltage, 3,000 V; ion transfer capillary temperature, 270 °C; source temperature 100 °C; sheath gas 18, auxillary gas 5 (arbitrary units); collision gas, argon; S-lens Voltage was optimized for individual metabolites; scan time of 50 millisec/transition and ion polarity positive. In case of azadirachtin (15.6 pg to 1 ng), nimbin (3.4 pg to 0.25 ng) and salanin (7.8 pg to 0.5 ng) on column was used to construct the standard curve. The overall scheme for the quantification of neem metabolites is shown in the Fig. 1. The UHPLC-MS/SRM chromatogram for lowest standard and metabolites from seed, standard curve for three metabolites and final quantification of neem metabolites from various tissues are illustrated. The final absolute quantification was done based on the constructed standard curve (ratio versus concentration) of individual metabolites.

Metabolites pathways analysis

The quantified metabolites data points from different explants or tissues were used to compare the gene expression pattern using WGCNA package in R (Langfelder & Horvath, 2008) for tracking genes involved in azadirachtin biosynthetic pathway. The clustering program in WGCNA provided the initial correlation with gene expression and metabolite concentrations in different tissues of neem tree (Fig. 2B) (Table S7). The annotation of genes from KEGG and BLAST were used to select the candidate genes in secondary metabolite biosynthetic pathway. We selected the genes with best bit score for each step in terpenoid biosynthetic pathway. The selected gene annotation was confirmed by BLAST results against NCBI-nr database. In case of multiple hit for pathway function, the expression levels of genes in different tissues were compared to match the metabolite concentration by considering developing endosperm and mature leaf as contrasting datasets for azadirachtin concentration. Pearson’s correlation value was also calculated between the expression values of each gene in tissues versus Azadirachtin concentration (Table S8A). Further genes involved in conversion of squaline to azadirachtin could not be assigned to specified function. However, the genes showing a high expression level in developing endosperm and having high Pearson’s correlation with azadirachtin concentration were hypothesized to be part of azadirachtin biosynthesis. The hypothesized set of annotated genes in each pathway is listed in Table S8 and also shown in heatmap pattern (Fig. 3) using R statistical program.

Figure 2 Gene expression and metabolite differences among different tissues of neem.

(A) Comparison of genes expressed commonly and uniquely in flower and bud, developing endosperm, mature leaf, mature fruit, fruit coat and pulp tissues. (B) Clustering dendrogram of samples based on gene expression values and corresponding metabolite concentration. The heat map shows the higher expression of the genes in specific tissue.

Figure 3 Proposed Azadirachtin biosynthetic pathway in A. indica.

Azadirachtin is the most prominent biopesticide belonging to organic molecule class called tetranortriterpenoids. Triterpenoids derived basically from squalene which in turn is derived from geranylpyrophosphate (GPP). GPP can be derived from mevalonate (MVA) or 2-C-methyl-D875 erythritol 4-phosphate (MEP) pathway. Enzymes of MVA pathway are as follows: AACT, acetyl-CoA acetyltransferase; HMGS, 3-hydroxy-3-methylglutaryl CoA synthase; HMGR, 3-hydroxy-3-methylglutaryl-coenzyme A reductase; MVK, mevalonate kinase; PMK, phosphomevalonate kinase; PMD, diphosphomevalonate decarboxylase. Enzymes of MEP pathway are DXS, 1-deoxy-D-xylulose-5-phosphate synthase; DXR, 1-deoxy-D xylulose-5-phosphate reducto-isomerase; MCT, 2-C-methyl-D-erythritol 4-phosphate cytidylyltransferase; CMK, 4-diphosphocytidyl-2-C-methyl-D-erythritol kinase; MDS, 2-C methyl-D-erythritol 2,4-cyclodiphosphate synthase; HDS, 4-hydroxy-3-methylbut-2-enyl diphosphate synthase; HDR, 4-hydroxy-3-methylbut-2-enyl diphosphate reductase. Isopentenyl pyrophosphate isomerase (IPPI) catalyzes the isomerisation of isopentenyl pyrophosphate (IPP) to dimethylallyl pyrophosphate (DMAPP), whereas conversion of IPP to geranyl pyrophosphate (GPP) is catalyzed by geranyl pyrophosphate synthase (GPS). GPP is further converted to farnesyl-diphosphate (FPP) and squalene by farnesyl-diphosphatesynthase (FPS) and squalene synthase (SS) respectively. Solid arrows indicate known steps, whereas broken arrows represents unknown intermediates and enzymes. Numbers beside coloured blocks indicate tissue types (1, developing endosperm; 2, mature leaf; 3, mature fruit; 4, seedling root; 5, fruit coat and pulp; 6, seedling shoot; 7, open flower and flower bud) and heatmap is represented as expression RPKM value.

Results

De novo sequencing and assembly of neem genome

We sequenced the neem genome using Illumina and 454 platforms. Illumina HiSeq paired-end (2 × 100 nts) sequencing yielded 13.86 Gb of high quality data for the neem Genotype 1 (GKVK, Bangalore). De novo assembly resulted in 216 Mb (version 1.1) with 21X coverage using Velvet software. This analysis produced a total of 94,780 scaffolds where 90% of the genome was covered by scaffolds length longer than 1,000 bp. The N50 was about 22 Kb with 31.13% of GC content and the longest scaffold length was 241 Kb. In addition to Illumina data, we also generated 1.13 Gb (2,762,254 reads) data using Roche 454 GS FLX + chemistry for the Genotype 1. The average read length was 410 nts and longest 454 read length was 1,596 nts. The 454 assembly was generated using MIRA program (Table 1). The draft genome quality was further improved through hybrid assembly by merging Illumina contigs and 454 contigs using CD-HIT-est by keeping minimum similarity cut-off of 90%. The total size of hybrid genome assembly for neem Genotype 1 was improved to 267 Mb. Assembly statistics of the improved assembly neem Genotype 1 is shown in Table 1. Clustering approach in hybrid assembly has significantly reduced the number of contigs from 94,780 to 68,604 and decreased the number of N’s from 2.22% (4,841,912 nts) to 1.81% (4,842,395 nts). Although clustering approach reduced the N50 from 22.3 Kb to 15.95 Kb, the longest and shortest contig length remained unchanged. Hence, the clustered hybrid assembly (version 2.1) was chosen for further detailed analyses. To confirm the completeness of neem genome assembly, we analyzed conserved eukaryotic genes (CEG) in the neem hybrid genome assembly. This analysis was able to identify 224 out of 248 complete CEGs in the neem assembly.

Neem genome annotation

Gene prediction was performed using two programs, Augustus and GenScan, which led to identification of 40,130 and 52,617 genes, respectively. In total, we identified 44,495 genes after merging and clustering of genes from these two programs. The genic region of neem genome was about 114 Mb (42.5%). The initial gene prediction was carried out without masking the repeat regions in the genome to avoid missing of simple sequence repeats in the coding regions and to predict proper exon/intron boundaries. To annotate the neem genes, annot8r program was used by incorporating functional proteins from GO, EC and KEGG and merged with BLAST results (Table S9). There were 29,050 unique genes in the neem genome that were free from any repeat elements.

We used RNAseq data to validate predicted neem genes (Table 2). This analysis revealed thousands of genes expressed in various neem tissues including flower (21,927 genes), mature fruit (17,366), developing endosperm (19,480), mature leaf (19,308), fruit coat and pulp (21,537), seedling root (20,798), and seedling shoot (20,199). The analysis also showed that 3,008 genes exhibited tissue specific expression profile, while 13,711 genes were expressed in all the tissues (Fig. 2A). We identified 80,867 transcripts (53 Mb) from de novo assembly of neem transcriptome using Trinity program. These transcripts were further used as supporting expression evidences for the genes that involved in metabolites biosynthesis pathways. The protein sequences for 44,495 genes were compared with proteome of 23 sequenced plant species. Of these, 23,125 genes (52%) were classified into 18,327 families (Fig. 4). Neem genome found to have 4,320 multi-gene families (Table S9).

Figure 4 Co-orthologous groups detected by Proteinortho using UPGMA method.

The clustering is based on similarity of the proteome among the plant species.

Gene ortholog analysis

Ancestral orthogroup reconstruction was done using Wagner Parsimony (Table S10) and likelihood based on birth–death model with equal gain-loss penalty (Table S11). The orthogroup reconstruction shows that 5,122 genes gained and 2,755 genes lost in comparison with ancestral nodes based on the gene family expansion in the neem genome.

Along with gene classification to orthogroups, the BLAST results of proteinortho (Lechner et al., 2011) showed 24,216 genes (54.42%) as common between neem and citrus (Fig. 5A). The comparative analyses revealed 20,279 and 21,931 genes that are unique to neem and citrus, respectively. Out of 20,279 unique genes, 5,832 genes were expressed in various neem tissues.

Figure 5 Comparison of neem genes with other sequenced plants.

(A) Schematic diagram showing common and unique genes between citrus and neem genomes. (B) Number and percentage of proteins having homolog hit in query (Arabidopsis, Populus, Grapes, Castor and Rice) with greater than 60% identity in genomes of neem and citrus.

Neem orthologous genes were compared with dicot (Arabidopsis, Populus and grapes) and monocot (rice) plants (Fig. 5B). 27,498 genes out of 44,495 genes were found to have orthologs (with more than 60% identity) in other plant species. Interestingly, 38% (16,997) of genes from neem did not show any orthologs in sequenced plant genomes. This analysis identified 16,997 genes that are unique to neem, which support the presence of unique gene families in neem tree. Among these unique genes, 680 genes share minor homology with hypothetical/predicted proteins in other plant species, while 2,343 were genes found to have no sequence similarity to either predicted or known genes. The list of unique genes in neem is summarized in the Table S12. The unique genes were further filtered for presence of repeat elements and expression evidences. More than 3,000 unique genes that have no repeat elements in the neem genes have expression evidence. This indicates that neem repeat derived genes are active in various neem tissues, which are interesting for future studies.

Repeat content in neem tree genome

De novo repeat identification in neem genome was performed using RepeatModeler (Smit & Hubley, 2008). This analysis identified 32.53% (87 Mb) of neem genome constitute for repeat elements, of which 17.06% of repeats were not annotated to any known repeat families (Table 3). The long terminal repeat (LTR), retro-transposons are the major classes of known repeats, which constitute about 10% of repeats the neem genome. The detailed repeat prediction is shown in Table S13. Our repeat prediction in neem (32.53%) is comparable to rice genome. Our RepeatModeler accuracy for repeat prediction in neem genome was further confirmed by independent analyses of the other two re-sequenced neem genomes (Genotype 2 = 27.63%, Genotype 3 = 26.76%). To verify repeats in the recently reported neem genome by Krishnan et al. (2012), we re-built the neem genome assembly reported by Krishnan et al. short read data (SRA1085705) using SOAPdenovo2 program. During our re-analysis of neem genome assembly from Krishnan et al. dataset (SRA1085705), we found 22 to 35% of genome harbour repeats (Table S5). In addition, we identified higher gaps with Ns (14.7% to 60%) for 31 to 37 k-mers from SRA1085705 (Table S5) as compared to neem genome assembly (1.81%) from our study.

Table 3 De novo repeat prediction from neem genome using Repeat Modeler program.

Repeat type	Subclass	Number of elements	Length in bp (%)	% of sequences	
RNA elements:					
SINEs:		119	10,033bp (0.01)	0.00	
LINEs:		2,554	124,0701 bp (0.46)	0.46	
	LINE1	1,918	100,2661bp (0.37)	0.37	
	LINE2	119	39,194 bp (0.01)	0.01	
LTR elements:		51,260	26,838,058 bp (10.02)	10.02	
	ERV_class1	462	130,859 bp (0.05)	0.05	
DNA elements:		17,356	7,112,032 bp (2.65)	2.65	
Unclassified:		16,6584	45,693,338 bp (17.06)	17.06	
Total interspersed repeats:			80,894,162 bp (30.20)	30.20	
Simple repeats:		43,430	1,601,548 bp (0.60)	0.60	
Low complexity:		99,976	5,150,192 bp (1.92)	1.92	

DNA polymorphism among neem genotypes

Simple sequence repeats (SSR), single-nucleotide polymorphisms (SNPs) and small insertions and deletions (InDels) are the most abundant DNA markers in any plant genomes. Three sequenced neem genotypes were compared to assess genome-wide genetic diversity. This analysis identified 140807, 108020 and 95840 SSRs from neem Genotype 1, Genotype 2 and Genotype 3, respectively (Tables S14A and S14B). Genes in Genotype 1 (2,217), Genotype 2 (1,665) and Genotype 3 (1,606) were associated with the presence of SSRs motif. SSR markers in the genic regions of Genotype 1 are summarized in Tables S14A and S14B. Tri-repeats were highest (1,841 SSRs) for genes as compared to other repeat units. The AAG/CTT (533), AG/CT (114) and A/T (127) were the most predominant SSR motifs in genic regions of neem genome (Tables S14B). The list of SSR associated genes in Genotype 1 has been provided in Table S15.

With the availability of three neem genomes, we developed an in-house analysis pipeline to predict in-silico polymorphism in SSR loci. Nearly 100 nts from upstream and downstream regions of SSR motif from Genotype 1 (reference) were used for SSR polymorphism analysis. These analyses resulted in 2,199 and 2,120 polymorphic SSRs in the Genotype 2 and Genotype 3, respectively. The concordance SSR analyses showed 571 SSRs were polymorphic among three neem genomes (Table S16). The in-silico identified polymorphic SSR markers can be used to test polymorphism in neem germplasm or natural plantation.

In addition, SNP and InDel were further analyzed among three neem genotypes using Genotype 1 as a reference genome. This analysis yielded 698,173 SNPs and 53,508 InDels for Genotype 2 and 860,215 SNPs and 66,171 InDels for Genotype 3, respectively. SNP annotation was carried out to discern presence of SNPs in the coding region of the genome. The SNP annotation showed that 80.70% and 79.16% of SNPs in Genotype 2 and 3, respectively, are located in the non-coding regions (upstream and downstream) of the genes. We identified 130,668 (2.78%) and 159,375 (3.21%) SNPs, for exonic regions of coding genes from Genotype 2 and 3, respectively. There were 31,312 and 37,864 SNPs in the coding sequence which represents the synonymous amino acid changes in protein coding regions of Genotype 2 and 3, respectively. There were 84,194 and 102,663 SNPs that showed non-synonymous amino acid substitutions in the protein coding regions of Genotype 2 and 3, respectively. The presence of SNPs in start and stop codons were less as expected. The details of SNP annotation is summarized in Table S17.

Azadirachtin biosynthesis pathways

We have quantified neem metabolites using high sensitive UHPLC-MS/SRM method, which can detect the picogram (pg) level of neem metabolites (azadirachtin—15.6 pg, nimbin—3.4 pg and salanin—7.8 pg). The concentration of neem metabolites in mature seeds (azadirachtin (11,046 pg/µg), nimbin (3,607 pg/µg) and salanin (5,235 pg/µg)) was higher than other neem tissues. The concentration of Azadirachtin was always higher in mature seeds, followed by developing endosperm, shoot, root, cambium, pulp, flower, bark and leaf. Nimbin concentration was higher in seed, followed by bark, cambium, endosperm, root, shoot, flower, pulp and leaf. Similarly for salanin, the trend was higher in seeds followed by endosperm, bark, cambium, root, shoot, leaf, pulp and flower. The quantification of neem metabolites from various tissues is shown in Fig. 1E. Three of these metabolites are always found to be higher in seeds than other tissues, therefore, we quantified these across various seed developmental stages (S1 = 10 days post seed setting, S2, S3 and S4 = 40 days post seed setting). The level of all these three metabolites was higher across developing seed stages (Fig. 6B).

Figure 6 Correlation of gene expression using qPCR and metabolites content in various neem tissues.

(A) qPCR performed with genes selected from RNA-seq data. Developing seeds were selected across different stages (S1—developing seed 1, S2—developing seed 2, S3—developing seed 3, S4—developing seed 4) based on growth stages show high gene expression as compared to other tissues. (B) LC-MS quantified metabolites data from various tissues.

Genes involved in various steps from tirucallol to azadirachtin are not yet established. Therefore, we devised bioinformatics approach by clustering expressed genes along with amount of metabolites (azadirachtin, nimbin and salanin) in each tissue using Weighted Correlation Network Analysis (WGCNA) method. This WGCNA clustering analysis identified azadirachtin biosynthesis genes that are up-regulated in developing endosperm, and had a minimal expression in other tissues such as leaf, flower, fruit coat and pulp (Table S8A). From this analysis, we identified more than 150 genes that were highly expressed in developing endosperm. However, the majority of these genes have homology to other plant species with a Pearson’s co-relation value above +0.8 for azadirachtin. To validate our method, we selected 10 genes with high correlation value (≥0.9) with azadirachtin (Table S8A) to perform quantitative PCR (qPCR). For this analysis, we used RNA from leaf, callus and developing seed (S1, S2, S3, S4 different stages of seed development) from neem Genotype 1. This analysis showed that 8 out of 10 genes (Fig. 6A) have high expression in developing seeds and low expression in other tissues (leaf and callus). The qPCR results concordance with WGCNA based clustering of RNA-seq and UHPLC-MS/SRM dataset. The highly expressed genes such as transketolases (Ai02g19151 and Ai02g23582) and dehydrogenases (Ai02g25309 and Ai02g12737) (Fig. 6A) were among the top ranked in Pearson’s co-relation value in WGCNA and qPCR analysis.

Organelle genomes assembly and annotation

We filtered reads which map to chloroplast and mitochondrial genomes of other plant species. The assembly and annotation statistics for organelle genome are shown in Table 4. The chloroplast genome assembly contains 60 scaffolds with size of 112,958 nts, which accounted for 72% of average plant chloroplast genome. The chloroplast genome had N50 of 2,125 nts, GC content of 38.07% and longest scaffold length of 8,435 nts. Further, gene prediction and assembly of chloroplast genome was done using DOGMA (Wyman, Jansen & Boore, 2004), which showed 77 unique genes in chloroplast genome of neem (Table S18).

Table 4 Mitochondria and chloroplast genomes assembly statistics.

Assembly parameters	Mitochondrion	Chloroplast	
Total number of reads	15,659,391	22,211,576	
k-mer	63	61	
Assembled genome size (bp)	266,430	112,958	
Total number of contigs	348	152	
N50 (bp)	1,490	2,125	
Maximum contig length (bp)	9,110	8,435	
Minimum contig length (bp)	125	121	
% of bases in contigs ≥1,000 bp	60.71	63.33	
GC content %	43.1	38.07	
No of genes predicted	39	77	

The mitochondrial genome of neem was also assembled (Table 4). N50 for mitochondrial genome had 1,490 nts, the sequence covered 266,430 nts of the genome in 348 scaffolds. The GC content of the mitochondrial genome was 43.10%. Gene prediction and annotation of the mitochondrial assembly was done using Mitofy (Alverson et al., 2010), which identified 39 genes out of 41 reported mitochondrial genes (Table S19).

Genome orthology and synteny analysis

Genome size and chromosomal architecture was not available for neem tree. Synteny analysis was performed to obtain conserved chromosomal blocks of neem genome and genes in a closely related plant species. Our analysis revealed that the neem genome is phylogentically related to citrus. Therefore, we used the citrus genome (nine citrus pseudomolecules) as a reference to order neem contigs. This comparative analysis anchored 24,902 neem contigs onto 9 citrus chromosomes (Fig. 7). This anchoring method ordered 161 Mb (62%) of neem genome covering, which is equivalent to 48% of citrus genome. Detailed anchoring analysis revealed that 497 syntenic blocks with 12,176 synteny hit with citrus genome (Fig. 7).

Figure 7 Schematic representation of syntenic relationship between citrus and neem.

The colored line (1–9, UNK) represents the syntenic blocks in neem anchored to chromosomal region in citrus. The turquoise colored blocks show the synteny of un-anchored neem scaffolds with citrus.

Discussion

Neem is medicinally, agriculturally and environmentally important tropical tree in Indian subcontinent. Neem is well-known for its complex tetranortriterpenoids compounds such as azadirachtin, nimbin and salanin, which are the main constituents in insecticidal and pharmaceutical formulations (Brahmachari, 2004). Unlike inorganic synthetic pesticides, neem compounds are best-known bio-pesticides, which can easily be degraded and have lower pesticidal toxicity in the environment. Biochemical compounds from neem tree have been well investigated in the 20th century (Brahmachari, 2004). However, genetic, molecular and genomic resources are not well developed to understand genes and biochemical pathways in neem. Recently, attempts were made to generate ESTs (Rajakani et al., 2014; Narnoliya et al., 2014) and genomic (Krishnan et al., 2012) resources. However, these studies have generated limited number of ESTs (Rajakani et al., 2014; Narnoliya et al., 2014) and also non-availability of genome assembly and genes (Krishnan et al., 2012) in the public domain. Our study aimed to develop comprehensive genomic, transcriptomic and metabolomic resources for neem tree.

We sequenced three neem genomes from three distinct geographical regions of southern India. We annotated the neem genome to the best of our knowledge using available bioinformatics tools. We could able to assemble 70% (267 Mb) of genome based on estimated genome neem size by Ohri, Bhargava & Chatterjee (2004). More than 90% of conserved eukaryotic genes (CEGs) mapped to de novo assembled neem genome. Our prediction of genes (44,495) and TE-related genes (35%) are comparable with well annotated rice genes (http://rice.plantbiology.msu.edu) (Kawahara et al., 2013). More than 30,000 genes in the neem genome are supported by expression evidences, 13,711 genes expressed in most of neem tissues and 3,000 genes expressed in a tissue specific manner.

Our comprehensive analysis predicted about 87 Mb (33%) repeats in the neem genome in contrast to previous study (Krishnan et al., 2012). They have reported that neem genome contains the lowest repeat content (13.03%) in the plant kingdom (Krishnan et al., 2012). The previously published neem genome by Krishnan and colleagues has not released genome assembly and hence we cannot really make use of information. We tried to rebuild the neem genome using Krishnan et al. short reads dataset (SRA1085705). This analysis predicted more than 20% of repeats and constitutes higher gaps with lots of Ns (up to 60%) (Table S5). According to our knowledge, they have underestimated the repeat content in the neem genome and their results are not reproducible (Table S5). Our assembly generated from neem Genotype 1 showed higher repeats (33%) content and lesser Ns (less than 2%) than the study by Krishnan and colleagues.

Other highlight of our study is that we sequenced three neem genotypes from varied environmental conditions, which assisted us to identify DNA molecular markers such as SSRs, SNPs and InDels. We obtained about 2.9 SNPs and 0.22 InDels per 1,000 nts on the reference neem genome (Genotype 1). SNP and InDel markers density distribution in neem genome are lower than citrus genome (3.6 SNPs and 0.6 InDels per Kb) (Xu et al., 2013). The lower SNP and InDel diversity in the neem genome and the genic region indicates that neem trees might be genetically less diverse. Neem trees might be forced to self pollinate because of bisexual nature of flowers, closed floral anatomy and lack of self incompatibility (Puri, 1999). Other reason might be due to the presence of insect repellent compounds in leaves and flowers, which may significantly reduce cross pollination. Molecular markers (SNPs, InDels and SSRs) from this study are highly useful in identification of elite genotypes, tagging of traits and cloning of genes involved in biochemical pathways in neem through genome-wide association studies.

In the molecular phylogeny, neem is genetically related to sweet orange (Citrus sinensis) at the family level (Xu et al., 2013). A lot more genetic and genomic resources are available for citrus plant (Wu et al., 2014) as compared to neem. Therefore we used citrus for comparative analysis with neem genome (Fig. 5A). We observed that extensive syntenic blocks (62% of neem genome) between neem and citrus chromosomes and about 50% (24,216) of neem genes were conserved in the citrus genome. Citrus has been well researched plant for limonoids (Xu et al., 2013; Wu et al., 2014), which are highly oxygenated limonoids present in both Rutaceae and Meliaceae. Molecular resources from our study will help in dissecting common and specific limonoid pathways in neem and citrus.

The proposed biosynthetic pathway of azadirachtin in neem is not well studied. The tirucallol (C30 triterpene) a steroid triterpenoid, is a possible precursor for azadirachtin biosynthesis in neem (Johnson, Morgan & Peiris, 1996; Ley, Denholm & Wood, 1993). In the first step, two molecules of farnesyl diphosphate combine to generate tirucallol molecule followed by losing three methyl groups and oxidized to form apotirucallol (a tetranortriterpenoid, or limonoid). Then it loses the four terminal carbons (Dewick, 2011; Ley, Denholm & Wood, 1993). The third ring of apotirucallol is oxidized to form the C-seco limonoids, nimbin and salannin (Johnson, Morgan & Peiris, 1996; Ley, Denholm & Wood, 1993; Puri, 2003; Saxena, 1989). These molecules are heavily oxidized and cyclised to produce azadirachtin (Aerts & Mordue, 1997; Hosfelt, 2008).

Our study added genomic, transcriptomic and metabolites data to support limonoids biosynthesis pathway genes in neem. To understand the neem metabolites bio-synthesis, we quantified major metabolites using a sensitive UHPLC-MS/SRM method. We identified the known secondary metabolite pathway genes including farnesyl diphosphate synthase (Ai02g10209), squalene synthase (Ai02g1634), geranyl diphosphate synthase (Ai02g31423), mevalonate kinase (Ai02g16520) and other unannotated genes upstream to squalene (Table S20). Across different stages of developing seed (10 to 40 days after seed setting), we observed increased content of major metabolites as compared to leaf and callus tissues (Fig. 6B). In mature seeds, the concentration of neem metabolites (azadiractin (11,046 pg/µg), nimbin (3,607 pg/µg) and salanin (5,235 pg/µg)) was higher than other neem tissues as reported earlier. Azadirachtin was 5,000 fold higher in seed as compared to leaves. Azadirachtin is the major complex limonoids in neem tree. Genes involved in various steps from tirucallol to azadirachtin are not yet established. Genes identified from this study will help to identify biochemical pathways in future. Clustering of genes expression and metabolites data aided in identification of genes that associate with azadirachtin biosynthesis pathways. The comprehensive resources from our study will help to unlock the biochemical pathways in neem.

Conclusion

Neem is an important tropical evergreen tree in India, used for centuries in agriculture and traditional medicine. In this study, we report the detailed analysis of neem genomes, transcriptomes and metabolites. We identified possible candidate genes involved in azadirachtin biosynthesis pathways. Genomic resources such as sequence of genomes, genes, transcripts, SSRs, SNPs and InDels from this study will have a profound application to study diversity, traits association, with biopesticidal properties and biochemical pathways in neem and other species of Melieacea family.

Supplemental Information

Table S1 Genome assembly (version 01) and annotation of neem genotypes using Illumina PE data

Click here for additional data file.

Table S2 Gene prediction statistics using Augustus and Genscan for Genoytpe 1 hybrid genome assembly (version 2)

Click here for additional data file.

Table S3 Neem genome annotation

Click here for additional data file.

Table S4 Comparison of de-novo repeats prediction in A. indica with P. trichocarpa, R. communis and A. thaliana using RepeatModeler program

Click here for additional data file.

Table S5 Assembly statistics of re-analysed Krishnan and co-workers data (SRP013453)

Click here for additional data file.

Table S6 List of primers used in qPCR analysis

Click here for additional data file.

Table S7 Metabolite quantity measurement in different explants of neem tree (Genotype 1)

Click here for additional data file.

Table S8 List of genes with very high expression in developing endosperm of neem and also with high correlation with nimbin content in the tissues under study

Click here for additional data file.

Table S9 Multigene family in the neem genome

Click here for additional data file.

Table S10 Ancestral orthogroup reconstruction wagner parsimony using equal gain-loss penalty, as implemented in the program count

Click here for additional data file.

Table S11 Ancestral orthogroup reconstruction using a birth–death model that allows for lineage specific gain/loss rates, as implemented in the program count

Click here for additional data file.

Table S12 Genes unique to neem with expression and without repeat elements

Click here for additional data file.

Table S13 Description of repeats prediction in A. indica using RepeatModeler program

Click here for additional data file.

Table S14 Simple Sequence Repeats (SSRs) prediction for genomes and genes from 3 Genotypes of A. indica and their comparison

Click here for additional data file.

Table S15 The list of SSRs in genes of neem Genotype 1

Click here for additional data file.

Table S16 The list of polymorphic SSRs among 3 genotypes

Click here for additional data file.

Table S17 Summary of SNP annotation for Genotype 2 and Genotype 3 by using Genotype 1 as a reference

Click here for additional data file.

Table S18 Genes annotated from neem chloroplast

Click here for additional data file.

Table S19 Genes annotated from neem mitochondria

Click here for additional data file.

Table S20 Neem secondary metabolites biosynthesis genes

Click here for additional data file.

Figure S1 Work flow of sequencing and genome assembly of neem genome

Click here for additional data file.

Figure S2 Schematic representations of GO classes in neem genome

Click here for additional data file.

We acknowledge Genomics facility (BT/PR3481/INF/22/140/2011) at Centre for Cellular and Molecular Platforms, Bangalore for sequencing of Neem genomes. We acknowledge Pradeep H, Aarati Karaba, Manojkumar S and Annapurna for their help in NGS library preparation and sequencing. We thank Ashmita G and Divya S for their help in manual curation of SSR markers. We are grateful to Rajanna, National Botanical Garden, University of Agricultural Sciences, GKVK campus, Bangalore for his help during neem sample collection.

Abbreviations

nts Nucleotides

UHPLC-MS/SRM Ultrahigh performance liquid chromatography-mass spectrometry-selected reaction monitoring

Kb Kilobases

RPKM reads per kilo base per million

WGCNA Weighted Correlation Network Analysis

SNP Single nucleotide polymorphism

SSR Simple sequence repeat

Additional Information and Declarations

Competing Interests

Author Contributions

DNA Deposition

The authors declare there are no competing interests.

Nagesh A. Kuravadi and Vijay Yenagi performed the experiments, analyzed the data, wrote the paper, prepared figures and/or tables.

Kannan Rangiah performed the experiments, analyzed the data, wrote the paper, prepared figures and/or tables, reviewed drafts of the paper.

HB Mahesh performed the experiments, analyzed the data, wrote the paper.

Anantharamanan Rajamani and Meghana D. Shirke analyzed the data, wrote the paper.

Heikham Russiachand analyzed the data, prepared figures and/or tables.

Ramya Malarini Loganathan, Chandana Shankara Lingu, Shilpa Siddappa and Aishwarya Ramamurthy performed the experiments.

BN Sathyanarayana performed the experiments, contributed reagents/materials/analysis tools, reviewed drafts of the paper.

Malali Gowda conceived and designed the experiments, contributed reagents/materials/analysis tools, wrote the paper, prepared figures and/or tables, reviewed drafts of the paper.

The following information was supplied regarding the deposition of DNA sequences:

1. Whole genome of neem deposited in NCBI accession number AMWY00000000.1 (Bioproject ID: PRJNA 176672)

2. Raw reads deposited in sequence read archive database (SRP052002).

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
