# Peer review of "Comprehensive analyses of genomes, transcriptomes and metabolites of neem tree"

_PeerJ, doi:10.7717/peerj.1066_

## Round 0.1 · original submission · Major Revisions

I have read the comments of the all reviewers -- three reviewers have major concerns about this MS. However, I see positives in this paper particularly for Neem research. Hence, I have decided to provide you with an opportunity to revise this article, particularly giving importance to published Neem genomic data and transcriptomes, and complete comparison of the published data on Neem.

Additionally, reviewers have pointed out several errors concerning writing and grammatical errors. I would recommend that the authors seek English editing services for the revised version or utilize a native speaker.

Please go through all the comments and provide replies in a constructive manner. The reviewers comments are critical for an improvement of your manuscript.

Reviewer 1 ·

Basic reporting

1. The article needs to be edited for the use of proper grammar. For example, there are many instances where definitive and indefinitive articles are missing. In the Abstract: “we sequenced whole genome of neem using Illumina” should be “we sequenced the whole genome of neem using Illumina”.

2. In the abstract: “The neem genome harbours ~32% of repeats”. Does that mean that 32% of the genome consists of repetitive sequence? Or does it mean that 32% of all repeats ever found in any genome can be found in the neem genome?

3. In the abstract: “We used an ultrahigh performance liquid chromatography-mass spectrometry-selected reaction monitoring (UHPLC-MS/SRM) to quantify metabolites…”. The indefinite article “an” implies that there is a noun coming up somewhere. I don’t see a noun to attach with the “an”.

4. In the author list, the name “aishwarya ramamurthy” is not capitalized? Is this person related to e.e. cummings? On the first page in the pdf provided for review, “aishwarya ramamurthy” has two names. On the has one name, “Aishwarya”. Is this person related to e.e. cummings or Elvis? Other people have initials at the end of their names “Sathyanarayana B N”, “Ramya Malarini L”, etc. List the authors’ names First(Given Name) Middle(or secondary names) Last(Family Name).

5. About 44,000 genes? Is the actual number in doubt? State the real number.

6. Line 33. The statement that genes involved in the azadirachtin pathway were annotated is a stretch. Genes were possibly found that are involved in the synthesis of squalene. However, squalene is a precursor to a very large number of metabolites. Were the genes involved in all of those pathways, therefore, also identified. Squalene is at least 7 steps away from azadirachtin. No claims can be made about finding genes in the azadirachtin pathway. Nobody will believe that assertion once they review the data.

7. Introduction, first sentence. “… native to Indian subcontinent.” Use the definite article: “… native to the Indian subcontinent”. There are many dozens of instances of this sloppiness.

8. Line 42. Missing period at end of sentence.

9. Line 45. Should be “have a reported cure”. There may be something to the compounds found in neem, but we are talking about unproven folk medicine. Asserting that these preparations and practices provide cures is too strong for a scientific journal.

10. Line 46. “agriculture, medicine and environment” should be “agriculture, medicine and the environment”. But even that is weak. I’m not sure what any sort of “application in the environment” would be. It may have environmental benefits over other types of pesticides.

11. Line 48. “chemical synthetic pesticides”. Maybe simply, “synthetic pesticides”.

12. Line 49. “from neem (Siddiqui 1942) and subsequently more than 150” should be “from neem (Siddiqui 1942), and subsequently, more than 150”.

13. I only three pages into this manuscript, and I am very frustrated with the poor grammar, punctuation and clarity. I will not offer additional corrections for grammar, punctuation and clarity. This manuscript must be submitted to a professional editing service. I will not serve as the editor.

14. Line 67. It has a limited utility?? That’s an understatement. If it is not public, in my mind, it is non-existant. You could word this more strongly. You could say that there has been a report of a previous genome assembly from neem, but the data remain privately held.

15. Line 73. “We have used the gene expression and metabolites data from various tissue to identify genes that are involved in azadirachtin biosynthetic pathway in neem.” There is a very clear standard that allows this claim to be made. We’ll see about that.

16. Line 79. To be clear. State whether or not leaves were collected from a single tree at each of the three locations.

17. Line 147. It is completely unclear how CD-HIT was used to combine the gene predictions from the two gene prediction programs. This is a huge issue. See point 31. I am not sure why CD-HIT was used, and I can imagine a number of ways that it could have been used badly. A lot more information is needed here.

18. The authors have not described how Augustus was trained.

18.5. It seems that transcript and protein evidence was not used when running Augustus. This is unfortunate. The gene predictions would have been more accurate if evidence had been allowed to guide Augustus.

19. Line 163. The combined transcriptome from 12 tissues was assembled using Trinity. What does this mean. Trinity was run 12 times, once with each RNA-seq data set from each tissue? Or, all 12 RNA-seq data sets were combined and Trinity was run once? If Trinity was run once, this was a really bad choice. You may have alternative splice forms that are from a single locus combined into a mixed-up hybrid transcript that doesn’t actually exist in neem.

20. Line 165. The consolidated genes were mapped with RNA-seq. That is badly written, but that is the theme here. Were RNA-seq reads mapped to just predicted gene sequences, predicted transcripts from predicted genes?? That is just wrong. Were the RNA-seq reads mapped to the genome? And then RPKM’s calculated for each gene? That would work.

21. Now I notice that the type of RNA-seq that had been performed is not described.

22. I really don’t understand why SSRs are included here. There are only three genotypes in this study. If the plants are diploid but nonetheless completely heterozygous, at most, each SSR locus would have six alleles. A shortlist of the most polymorphic sites was create? Why? The most polymorphic sites here are almost certainly not representative of the diversity of these sites within the whole species.

23. Line 212. SSR => SNP

24. Very nice detail in the UHPLC-MS/SRM methodology.

25. The numbering of figures (and therefore, I assume, of tables) is not properly ordered in the text.

26. Line 325. These CEGMA results are for complete or partial genes. Both numbers should be listed.

27. Line 332. What was the number of non-TE genes that had transcript support?

28. Line 166. Are there parameters for SeqMap? Were reads mapped uniquely or not? What degree of misalignment was allowed. What version was used? Was the 2008 version really used? There are much newer versions available.

29. Line 341. You don’t mean that the repeats are comparable which suggests that the types of repeats are similar. You are trying to indicate that the percentages of TE-genes are similar.

30. I wonder how complete the predicted gene set is. This can be tested by separately aligning the proteomes of one or two species to the genome and then also to the predicted neem proteins. If the number of alignments to the genome and proteome are identical, then probably all of the genes have been found. If more alternate species proteins have aligned to the genome compared to the predicted protein sequences, then some percentage of genes have likely been missed by the gene prediction programs (or there are errors in the predicted genes). This would be very useful and relatively easy analysis. This is all motivated because the number of multi-gene families in neem seems quite low to me. Is this low number due to the method of predicted gene clustering that I had questioned in point 17? Were all predicted genes pooled and then examined by CD-HIT?? Only predicted genes from the same locus should have been compared not all predicted genes from all loci.

31. Line 360. You mean that the entire 44,000 gene set, including TE-related genes, was compared to citrus? Did citrus also have TE-related genes? Are the 24,216 common genes all TE-related? How many are non-TE-related?

32. Line 361. The numbers are also meaningless until question 31 is resolved.

33. Line 366. The authors should look at how their ortholog program works. Often ortholog groups will contain groups of full-length proteins. Shorter protein sequences will not be able to be associated with full-length proteins due to the requirements of the software. Therefore, if neem contains a lot of partial genes sequences, many predicted proteins may be too short to be assigned to their proper ortholog group.

34. Line 367. You cannot claim that these are unique genes just because the ortholog analysis did not place them into orthogroups.

35. 2,343 may be a reasonable number of unique genes. I prefer the lack of blast support as evidence for the uniqueness of these sequences.

36. Line 387. I really do not understand how the accuracy of repeat prediction was confirmed by analysis of repeats in other genomes. This claim is based on the percent of repeats in these other genomes? Nobody will buy that. Additionally, the percentages reported in the text do not match the percentages reported in Supp Table 4.

37. Line 391. Are the SINEs, LINEs etc provided somewhere? Is there a table that breaks down the numbers for these various repeat types?

38. Line 421. Please explain further why low SNP/indel diversity would be explained by poor outcrossing. Are you implying that the plants are mostly inbred? As far as toxic flowers, your own Figure 1 suggests that the flowers are not that toxic. It seems as if mostly the fruits and seeds are protected.

39. Line 436. Supp Table 18. These are genes that have expression that correlates with metabolite levels. Correlation is only correlation. There are hundreds of genes listed here. Are you asserting that all of them are candidates for azadirachtin A, nimbin and salanin biosynthetic genes? It seems so. Is there no other way to trim this list? In the text, you highlight an apparent pollen allergen as having the best correlation. Is that the best possible example that I could have. Do a lot of these genes also correspond to seed storage proteins? Some more thought could be put into recreating this list to consist of genes that would not be dismissed out-of-hand as not being azadirachtin A, nimbin and salanin biosynthetic genes

40. Line 469. 1,12,958 nts. I don’t know what that means.

41. Line 482. Figure 7 is not mentioned anywhere in this section of the dicussion.

42. Line 485. Neem is closest to citrus as expected from phylogeny. Have you cited this phlogeny elsewhere? Cite it here.

43. Line 491. I like Supp Figure 3. Take out the arabidopsis and the neem unknown chromosome. Show the main neem chromosomes and the citrus chromosomes. Then add to citrus some other banding that indicates the position of the neem unincorporated contig alignments. Having that unknown chromosome in the synteny figure doesn’t add anything because it you aren’t showing synteny between the completely random unknown chromosome and citrus. Make these changes, and make this a main figure.

44. Line 504. Which molecular dendrogram?

45. Line 521. What did Krishnan et al actually predict? Where might they have gone wrong? What is the Krishnan et al paper? Year?

46. Line 524. Did Krishnan et al include TE-related genes in their predicted gene set? Why might they have underestimated the number of genes? I am not so concerned if you don’t like the Krishnan et al genome assembly. Make the comparisons and let me decide.

47. Line 532. I missed where you justified the qPCR genes as being involved in the azadirachtin pathway.

48. Line 535. Not one of these genes is a biosynthetic gene. One is an allergen, one is a transcription factor, and one is involved in the pentose phosphate pathway. They apparently have nothing to do with azadirachtin synthesis. I don’t see the point of mentioning this result.

49. Line 556. Discussion of the final steps in the biosynthesis of azadirachtin is kind of empty. You have not candidate genes for these final steps in the pathway. I have not read Heasley’s hypothesis on limonoid synthesis, but you really cannot have supported it. You only predicted genes that are possibly involved up to the synthesis of squalene. You do not propose even a single gene that could be involved in the limonoid pathway.

Experimental design

This is silly. I will not break down my review in this arbitrary fashion.

Validity of the findings

This is silly. I will not break down my review in this arbitrary fashion.

Additional comments

I have reviewed the manuscript by Yenagi et al. This manuscript is not close to being acceptable for publication. I have three general points to support this. 1. The writing and formating of the manuscript are terrible. The manuscript needs to be prepared by a professional editor before it could be resubmitted. 2. There are a number of technical questions that I have about the work. While the assembly of the neem genome looks good, there are additional steps that could be taken to provide additional support for the quality of the assembly. I also have a number of questions about some of the analyses that have been performed. Generally, without additional information, it is not possible to fully judge the quality of the analyses of the predicted genes and proteins. 3. The authors are particularly interested in three compounds, azadirachtin A, nimbin and salanin. These compounds are mentioned throughout the text. One apparent goal of the project was to find genes involved in the synthesis of these compounds. However, not one piece of credible data is provided that relates any predicted genes to these compounds. In addition to these three major criticisms, I have a list of over 40 particular points of concern that are provided in my review, and this does not include what must be hundreds of issues of poor grammar and punctuation.

Reviewer 2 ·

Basic reporting

Yenagi et al. have presented work on genomes, transcriptomes and metabolemes of a plant species. Although the plant species is important, their work lack scientific rigor, has methodological flaws, merely repeats partially what already has been published and has a misleading title. Therefore, it cannot be published in its current form.

I would recommend that the authors change the manuscript title, and present data only from metabolites and the transcriptomes that have not been sequenced before, as a new manuscript. Additionally, all the scripts used to generate assembly and along with the versions of all tools used for analyses so that their data can be reproduced.

Experimental design

Major and compulsory corrections:
1. As per the authors, raw data is submitted with the accession number SRP052002. However, data under this accession is not public. The authors must submit all the raw data.
2. Bad assembly like what the authors have presented with N50 of 22kbp can have serious implications on the downstream analyses and results? The authors must improve their N50 if they want genome assembly comparison with the previously sequenced genome to be published.
3. Since there are multiple genome assemblies mentioned in this study, the authors should mention clearly which particular assembly was used for each analysis to avoid confusion.
4. Critical paper, for example Rajakani et al. (Tree Genetics and Genomes, 2014), is not mentioned by the authors, which the paper must.
5. What were the training sets used for training Augustus and GenScan? What are the exact numbers of Ns obtained from reassembly of data from Krishnan et al?
6. The authors state, “Neem genome sequencing has been attempted previously….”. They also state, “it has limitation in utility because the genome data is not available to public”. We checked the earlier Krishnan et al 2012 paper, the underlying submitted data and found the authors’ statement very misleading. Both the raw data and the scripts used to generate assembly along with the versions are available for the Krishnan et al. 2012 paper in the public domain and as a part of the files submitted for the paper. Anyone who is skilled in bioinformatics and computational biology can generate scaffolds from the raw data using the scripts submitted as a part of that paper.
7. As the assembly algorithm changes, so does the genome assembly quality. Krishnan et al. used a previous generation of the assembly tool, therefore, the authors must use the raw reads from Krishnan et al 2012 paper along with SOAPdenovo2 and then perform comparison with their current assembly.
8. N50 in supplementary Table 20 is wrongly listed as 740 bp. As per the previously published genome of the same tree, the scaffold N50 is 452,028 bp. Therefore, the authors’ assembly is worse than the previously published information. How can the authors plan to publish worse assembly?
9. If the authors are interested in the genome work, they should perform genome comparison and not report genome sequence, which is already been done.
10. How are the gene numbers obtained? By counting the actual assemblies, or counting the unique component IDs?
11. Fig 3 is a repeat of what was already been tried before. Is there anything new in the phylogenetic map that was not known or published before?
12. Fig 6A – Fold change compared to what? What was the control?
13. RNA-seq library and assembly details are missing.
14. Figure 5 (big heatmap) – what are the genes/transcripts whose RPKMs are plotted?
15. The assembly available for download from Bioproject ID: PRJNA 176672 does not match the statistics of the final hybrid assembly. Longest contig = 47608bp, and has 125535 sequences. Why?
16. Mitochondrion and chloroplast assemblies are not available for download.
17. The number of genes in the combined tissue transcriptome (~32000) would be expected to be much higher than the individual tissue transcriptomes (20000) in the Krishnan et al study. The authors themselves listed the number of genes expressed per tissue in figure 1a, to be close to 20000. This, again, is scientifically incorrect and misleading.
18. For merging gene predictions by Augustus and GenScan, running BLAT was redundant since CD-HIT-est has the capability of clustering and merging similar predictions.
19. The authors must mention details on “How were variants called”?
20. Why did the authors identify genes upregulated specifically in the endosperm? Are anything known regarding the plant’s biosynthetic pathways being upregulated in the developing endosperm?

Minor Points:
1. The paper does not read well generally, and needs a lot of language editing. There are many grammatical and sentence construction errors everywhere alongside missing citations. The entire paper needs to be re-written.
2. Figure 4 image quality is very poor.
3. Figure 4 – reference for chemical structures missing.
4. Figure 3b legend is missing.
5. Read QC statistics are not summarized.
6. Method details for MIRA assembly of 454 reads are not mentioned.
7. List the sources (and versions, if applicable) of all reference proteomes and genomes of other plant species downloaded for comparison purposes are missing.

Validity of the findings

Data presented in the study is not robust and controlled. The conclusions drawn are not acceptable.

Additional comments

If the authors are interested in the genome work, they should perform genome comparison with the new version of the tool but with reads from published assembly and then compare genomes.

Reviewer 3 ·

Basic reporting

In this manuscript, the authors presented a regourous analysis of the draft genome of Neem (Azadirachta indica),a widely available and extensively used tropical Tree from Indian subcontinent. This draft genome and transcriptome promises to be a useful resource for understanding many important neem traits for biomedical and other applications.
The article is well written, and the topic, general methods, and question are well suited to PeerJ and draws wider scope of audience.

Experimental design

Experiments are well planned and presented.

Validity of the findings

Overall, the manuscript is well organized and present an sequence of important Tropical Tree Neem by up-to-date genomic and metabolomic methods and warrants its acceptance for publication.

Additional comments

It is desirable to provide the parameters of the tools and scripts used on to Github or as supplemental files.

Reviewer 4 ·

Basic reporting

It is strange to see the poor quality of figures in the MS. Such type of papers should have circus for comparison of genome feature, including others such as heat map. The lack of good figures lowers the quality of MS also.

Experimental design

Line No. 257-259: The author says that “There are no commercially available internal standards for neem metabolites, therefore, we used estrone-d4 (Steraloids Inc, USA) as an internal standard to construct standard curve and for quantification.” How much reliable is this method of screening in absence proper protocol.

Validity of the findings

Line No. 35-36: Why authors have indicated that identified SNPs can only be used for genetic diversity?. Is there any specificity with these SNPs that these can’t be used for other applications?.

Did authors validation of identified SSRs in wetlab? if not ...why not

Additional comments

Comment #1: Is there any specific reason in writing the name of author “aishwarya ramamurthy” in different manner in the first page. I feel it should be “Aishwarya Ramamurthy”.
Comment #2: This paper claims to sequence genome of Neem but genome sequence paper of this tree is already been published in 2012 (http://www.biomedcentral.com/1471-2164/13/464, A draft of the genome and four transcriptomes of a medicinal and pesticidal angiosperm Azadirachta indica. Neeraja M Krishnan et al. 2012, BMC Genomics, BMC Genomics 2012, 13:464).
Comment #3: Line No. 35-36: Why authors have indicated that identified SNPs can only be used for genetic diversity?. Is there any specificity with these SNPs that these can’t be used for other applications?.

Comment #4: Line No. 257-259: The author says that “There are no commercially available internal standards for neem metabolites, therefore, we used estrone-d4 (Steraloids Inc, USA) as an internal standard to construct standard curve and for quantification.” How much reliable is this method of screening in absence proper protocol.

Comment #5: Did authors validation of identified SSRs in wetlab?

Comment #6: The figures 1, 2, 3 and 4 are not of good quality….there is lot of scope for improvement of these figures.

Comment #7: It is strange to see the poor quality of figures in the MS. Such type of papers should have circus for comparison of genome feature, including others such as heat map.

---

## Round 0.2 · accepted · Accept

Dear Authors,

This manuscript is now accepted.

Reviewer 3 ·

Basic reporting

Resubmitted manuscript is improved in quality of english.

Experimental design

"No Comments"

Validity of the findings

"No Comments"

Additional comments

Revised manuscript is of good quality and addressed the referees concerns.